# Bioguided Fractionation of Local Plants against Matrix Metalloproteinase9 and Its Cytotoxicity against Breast Cancer Cell Models: In Silico and In Vitro Study

**DOI:** 10.3390/molecules25204691

**Published:** 2020-10-14

**Authors:** Maywan Hariono, Rollando Rollando, Jasson Karamoy, Pandu Hariyono, M. Atmono, Maria Djohan, Wiwy Wiwy, Rina Nuwarda, Christopher Kurniawan, Nurul Salin, Habibah Wahab

**Affiliations:** 1Drug Discovery Research Group, Faculty of Pharmacy, Sanata Dharma University, Campus III, Paingan, Maguwoharjo, Sleman 55282, Indonesia; jassonkaramoy70@gmail.com (J.K.); michaelpandu99@gmail.com (P.H.); tryatmononew@gmail.com (M.A.); mariangelinadj@gmail.com (M.D.); w1wy.leo@gmail.com (W.W.); 2Pharmacy Program, Faculty of Science and Technology, Ma Chung University, Malang 65151, Indonesia; ro.llando@machung.ac.id (R.R.); christopher.d.k555@gmail.com (C.K.); 3Faculty of Pharmacy, Padjadjaran University, Jatinangor, Sumedang 45363, Indonesia; rina.nuwarda@unpad.ac.id; 4Malaysian Institute of Pharmaceuticals and Nutraceuticals, National Institute of Biotechnology Malaysia, Halaman Bukit Gambir, Bayan Lepas 11900, Malaysia; hanim@nibm.my; 5Pharmaceutical Technology Department, School of Pharmaceutical Sciences and USM-RIKEN Centre for Ageing Science (URICAS), Universiti Sains Malaysia, Minden 11800, Malaysia; habibahw@usm.my

**Keywords:** MMP9, PEX9, cancer, bioguided, fractionation, ageratum, screening, in silico, in vitro

## Abstract

Matrix metalloproteinase9 (MMP9) is known to be highly expressed during metastatic cancer where most known potential inhibitors failed in the clinical trials. This study aims to select local plants in our state, as anti-breast cancer agent with hemopexin-like domain of MMP9 (PEX9) as the selective protein target. In silico screening for PEX9 inhibitors was performed from our in house-natural compound database to identify the plants. The selected plants were extracted using methanol and then a step-by-step in vitro screening against MMP9 was performed from its crude extract, partitions until fractions using FRET-based assay. The partitions were obtained by performing liquid–liquid extraction on the methanol extract using *n*-hexane, ethylacetate, *n*-butanol, and water representing nonpolar to polar solvents. The fractions were made from the selected partition, which demonstrated the best inhibition percentage toward MMP9, using column chromatography. Of the 200 compounds screened, 20 compounds that scored the binding affinity −11.2 to −8.1 kcal/mol toward PEX9 were selected as top hits. The binding of these hits were thoroughly investigated and linked to the plants which they were reported to be isolated from. Six of the eight crude extracts demonstrated inhibition toward MMP9 with the IC_50_ 24 to 823 µg/mL. The partitions (1 mg/mL) of *Ageratum conyzoides* aerial parts and *Ixora coccinea* leaves showed inhibition 94% and 96%, whereas their fractions showed IC_50_ 43 and 116 µg/mL, respectively toward MMP9. Using MTT assay, the crude extract of Ageratum exhibited IC_50_ 22 and 229 µg/mL against 4T1 and T47D cell proliferations, respectively with a high safety index concluding its potential anti-breast cancer from herbal.

## 1. Introduction

Cancer cases were recently diagnosed in approximately 1,806,590 cases, which is the equivalent of approximately 4950 new cases each day. In addition, there will be approximately 48,530 new cases of ductal carcinoma in situ of the breast diagnosed in women [1]. It is also the second leading cause of cancer-related deaths among women worldwide [2]. Breast cancer is classified into four major molecular subtypes based on a few biomarkers such as hormone receptors (HRs), human epidermal growth factor receptor 2 (HER2), and/or extra copies of the HER2 gene [3]. The subtypes i.e., luminal A (HR+/HER2−), luminal B (HR+/HER2+), HER2+, and triple-negative (estrogen receptor, progesterone and HER2-), have each different risk factors for incidence, disease progression, preferential organ sites of metastases, and therapeutic response [4]. Although surgery and current chemotherapy have shown positive treatment outcome for breast cancers, more effective drugs are still urgently needed to improve the drug selectivity and to overcome drug-resistance [5,6]. The urgency is compounded by the fact that only luminal A, luminal B, and HER2+ subtypes can be targeted by drugs such as tamoxifen and trastuzumab [7]. Unfortunately, there is no drug found suitable for the triple-negative type, although drugs such as carboplatin, paclitaxel, and doxorubicin are currently undergoing clinical trials for triple-negative breast cancer in Canada [8].

Studies have shown that high expression of matrix metalloproteinase9 (MMP9) occurred in breast cancer cells. MMP9 is a proteolytic enzyme belonging to MMP superfamily that lowers the value of the extracellular matrix (ECM) causing angiogenesis and cancer cell migration [9,10]. The activity of the enzyme is contributed by the zinc ion which interacts with the histidine triad in the catalytic site [11]. MMP9 was shown to be differentially expressed within different molecular subtypes of breast cancer. Overexpression of MMP9 is a clear feature of triple-negative and HER2+ breast cancers [12] and studies have suggested that MMP9 appears to be an important target for the triple-negative subtype of breast cancer [13], as beside overexpression, it is also involved in extracellular-matrix remodeling, plays a direct role in the expansion of tumor cells, and promotes tumor metastasis [10].

However, despite many efforts over many years to develop synthetic MMP inhibitors, all MMP inhibitors have failed in the clinical trial stages because of the onset of significant dose-limiting musculoskeletal toxicity or lack of efficacy [14]. These compounds have either a hydroxamate functional group or related chelator chemistry that acts to perturb the critical coordinating zinc in the catalytic domain, which results in the loss of enzymatic activity. Unfortunately, the catalytic site of MMP9 is highly conserved in the whole of MMP’s family which results in non-selective inhibitions by those compounds [15]. Thus, an alternative mechanism for targeting MMP9 in which the catalytic zinc is not targeted has been proposed to overcome the selectivity issue [16]. Example of such an alternative is targeting the hemopexin-like domain (PEX9) of MMP9 instead of the catalytic domain [17]. PEX9 contains non-conserved amino acid residues in its binding pocket [18] and is located next to the catalytic site [19], thus targeting PEX9 might offer as a good strategy for the development of selective inhibitors for the cancer treatment [20,21].

The use of natural product in cancer therapy has been attracting cancer survivors to choose the material that they believe safer than the synthetic drugs [22]. This prompts so many researches on natural product anticancer discovery to prove its scientific medicinal effect. For example, Bacopaside (bac) I and II, which are triterpene saponins isolated from *Bacopa monnieri*, significantly reduce the cell migration and its apoptosis, inducing combination of triple negative cancer cell lines (MDA-MB-231) [23]. Another scientific proof on anticancer activity from natural product was also shown by *Elaeagnus angustifolia* plant extract, which inhibits epithelial-mesenchymal transition and induces apoptosis via HER2 inactivation and JNK pathway in HER2+ breast cancer cells [24]. Recently, anthocyanins, which are purified from *Vitis coignetiae*, enhances cisplatin sensitivity in MCF-7 human breast cancer cells through inhibition of Akt and NF-κB activation, strongly supports the use of herbal remedies in cancer therapy [25].

In this present study, virtual screening against PEX9 was performed as the protein target to shortlist compounds from natural product database potential in inhibiting MMP9 using in silico molecular docking. It is believed that the natural product compounds have unique structures (more chiral carbon) [26] essential for further optimization. The shortlisted compounds would then be correlated to their presence in the local plants. Eight plants were extracted using methanol and further tested for their inhibition against MMP9 in vitro. From this result, four active extracts were partitioned using four different solvents and then further in vitro screened for its activity against MMP9. The most active partitions were then subjected to fractionation using column chromatography. The collected fractions were then tested again against MMP9 and the active fraction was figured out for their chromatography-mass profile using GC-MS. To support the potency as anti-breast cancer, four active extracts were further evaluated for their inhibition against 4T1 cell, a triple-negative breast cancer cell model. For comparison to the luminal A cancer cell type, the extract was also tested to T47D breast cancer cell model. The flow chart of the studies is illustrated in Figure 1.

## 2. Results

### 2.1. Control Docking and External Validation

The control docking was carried out to assure that the parameters used in the docking simulation were correct. The results for the control docking showed the root mean square deviation (RMSD) between the docked and the crystal poses of the sulfate ion is 1.99 Å. Although the RMSD value is considered high for such a small compound, however, it is still within the prescribed 2 Å according to the literature [27]. The external validation was also carried out with a series of compounds previously tested for PEX9 inhibition experimentally to increase the confidence that the protocols used are appropriate. Figure 2a showed the overlapped docked poses of 17 ligands used in the external validation with the inset showed the superimposition of the docked and initial (crystallographic) poses of the sulfate ion. A blue shift in proMMP-9 tryptophan fluorescence was monitored to determine the binding affinity in dissociation constant (K_d_) of all derivative compounds as previously described. The K_d_ was determined using the Prism software package (GraphPad V5) to fit the data to Equation: ΔF/ΔF_Max_ in which ΔF is the nM shift for a given titration and ΔF_Max_ is the maximal nM shift observed overall [28,29]. The activities of these compounds can be defined as active (Kd < 1.00 µM), marginally active (1.00 < K_d_ < 1.50 µM), or inactive (K_d_ > 1.50 µM) [30]. Based on this definition, it is noted that 11 of them are active, three compounds are marginally active, and another three are inactive. Nonetheless, based on the binding affinity, all the 17 compounds are predicted to be active because their calculated binding affinity have negative values and are lower than that of the sulfate ion (−3.5 kcal/mol). Thus, when the calculated binding affinity was compared with the experimental K_d_ values, the results can be categorized into true positive (TP) for active compound predicted active, and false positive (FP) for inactive compound predicted active as presented in Appendix A). In this case, neither true negative (TN) for inactive compound predicted inactive nor false negative (FN) for active compound predicted inactive, could be applied. The validity of the control docking parameters was evaluated by the True Positive Rate (TPR). In machine learning, the true positive rate, also referred to sensitivity or recall, is used to measure the percentage of actual positives which are correctly identified [31]. The TPR was obtained by dividing the number of TP (11) with the total compounds (17) and then multiplied by 100. The results showed that the TPR was equal to 65%, indicate that the docking protocol is able to validate the activity of the published compounds. Therefore, based on both control docking and external validations, the parameters of docking were used for further virtual screening.

### 2.2. Virtual Screening

After validation, the docking protocol was then used to screen our in-house database that contained 200 natural compounds. The binding affinity was found to be contributed by the molecular interactions such as steric interaction, hydrophobic interaction, and hydrogen bonding. Table 1 presents the top 20 ligands with the lowest binding affinity values related with their affinities toward PEX9. The top 20 ligands show the binding affinity in the range of −11.2 to −8.1 kcal/mol demonstrating their ability to bind the binding pocket of PEX9 which is stronger than sulfate ion (control ligand). Figure 2b illustrates the superimposition of the docked poses of the top 20 ligands at the binding pocket of PEX9. The amino acid residues involved in the molecular interaction with the ligands are ALA13, GLU14, GLY16, VAL58, PHE59, GLU60, PRO62, LYS65, ARG106, and GLN154.

### 2.3. Plants Extraction

Eight plants were selected for further extraction according to their availability in our surrounding areas. These were *Cordyline fruticosa* (palm lily) leaves, *Amaranthus spinosus* (spiny amaranth) aerial part, *Turnera diffusa* (damiana) leaves, *Hibiscus rosa-sinensis* (china rose) leaves, *Ageratum conyzoides* (goat weed) aerial part, *Ixora coccinea* (jungle flame) leaves, *Plumeria alba* (white frangipani) leaves, and *Melaleuca leucadendron* (cayuput) leaves. The yields of these extracts are presented in Table 2. The condition of the extract was preserved so that no fungus or color changing occurred during the storage as well as testing.

### 2.4. Bioguided Fractionation against MMP9

The eight crude extracts were then tested for their inhibitions toward MMP9 using the in vitro Fluorescence Resonance Energy Transfer (FRET)-based assay. In the first screening, the concentration used was 1000 µg/mL. This is considered as the highest concentration in which crude extract was still soluble in the solvent used. Six of the eight crude extracts demonstrated more than 50% inhibition toward MMP9. Only extracts from Cordyline and Melaleuca showed negative value for its % inhibition. This could be due to the fluorescence background of the extract itself or possibly the extracts triggered the MMP9 activity, thus negate the needs for further investigation.

The highest inhibition was performed by Ixora (86%), Plumeria (85%), Amaranthus (81%), and Ageratum (75%). Both Hibiscus and Turnera only showed 55% inhibition. However, following the IC_50_ determination (Table 2), the most potent extract is Plumeria (24 µg/mL) followed by Ageratum (64 µg/mL), and Ixora (82 µg/mL). While, the other three extracts that showed IC_50_ > 100 µg/mL are Turnera (495 µg/mL), Amaranthus (783 µg/mL), and Hibiscus (822 µg/mL). It can be seen that although Amaranthus show high percentage of inhibition, its IC_50_ is considered as moderate. This might be due to the weaker binding of the compounds present at the PEX9 site, thus higher concentration (up to 1 mg/mL) was required to reach for at least 50% inhibition.

According to the MMP9 inhibition performed by the extracts, top four extracts with the IC_50_ values less than 500 µg/mL were partitioned using four different solvents and seeded for further in vitro assay against MMP9. The in vitro results of these partitions were presented in Figure 3 in which most of Ageratum and Ixora partitions show higher inhibition than Plumeria and Turnera. Therefore, in the next step, the best partitions of Ageratum (*n*-hexane) and Ixora (ethyl acetate) were subjected to the fractionation using column chromatography.

From n-hexane partition of Ageratum, on the one hand, four fractions were collected. On the other hand, ethyl acetate partition of Ixora was fractionated into three fractions. One fraction for each partition with the highest yields (fraction 2 of Ageratum and fraction 1 of Ixora; Appendix A) was selected for the final evaluation of in vitro MMP9 inhibition as presented in Figure 4. From the results, it was observed that both Ageratum and Ixora fractions performed high inhibitions against MMP9, IC_50_ 43 ± 60 µg/ mL and 116 ± 18 µg/ mL, respectively. Therefore, it could be shortly concluded that the bioguided assay against MMP9 has been successfully performed on the herbal samples which was initiated by in silico screening work.

### 2.5. In Vitro Cytotoxicity Assay

To support the in vitro enzymatic results, four top active extracts against MMP9 were further evaluated for their capability to inhibit 4T1, a breast cancer cell model having triple-negative type from mouse. This cell has been known to express MMP9 during breast cancer. The results of in vitro cytotoxicity assay are tabulated in Table 3. From the table, Ageratum extract is identified as the most active sample to inhibit the proliferation of 4T1 cell with IC_50_ 22 µg/mL. Interestingly, the IC_50_ of Ageratum extract was lower than the positive control, doxorubicin (IC_50_ = 37 µg/mL) describing the better potency of Ageratum extract than doxorubicin as antimetastatic agent. The other three extracts, i.e., the leaves of Ixora, Turnera, and Plumeria demonstrated moderate activities against 4T1 cell growth with IC_50_ 270, 104, and 151 µg/mL. The morphologies of 4T1 cells before and after the treatment with the four most active extracts are illustrated in Figure 5.

As illustrated in Figure 5a, untreated 4T1 cells showed large cell shape with firm cell nucleus, whereas the 4T1 cells treated with the extracts (Figure 5b–e) resulted in a decrease of living cell populations compared to the untreated cells. The cells appeared round and fragmented indicating a change in the cell morphology. The result indicated that Ageratum was cytotoxic to 4T1 breast cancer cells and has the potential as an anti-proliferative agent. However, further investigation is needed to confirm that these changes that caused the cell death are due to the process of either necrosis, apoptosis, or the proliferation inhibition.

In the T47D cytotoxicity evaluation, the highest inhibition was also demonstrated by Ageratum extract with its IC_50_ 163 µg/mL, followed by Plumeria (IC_50_ 229 µg/mL), meanwhile Turnera and Ixora showed lower inhibition activities with their IC_50_, respectively 1295 µg/mL and 2200 µg/mL. To determine the safety index, cytotoxicity assay on the non-tumorigenic cell line was also conducted. The results showed that the IC_50_ of Ageratum, Plumeria, Turnera, and Ixora against Vero cell were, respectively, 307 µg/mL, 225 µg/mL, 613 µg/mL, and 653 µg/mL. The safety index is calculated by dividing the IC_50_ of extract in the Vero cell by the IC_50_ of extract in individual cancer cell as presented in Table 3. As shown, the safety index of Ageratum is the highest value among other extracts in both 4T1 (SI = 14.27) and T47D (SI = 1.88) indicating the least toxic of the extracts to the cells. The drug-dose response curve of Ageratum extract against 4T1, T47D, and Vero cells was presented in Figure 6.

### 2.6. TLC and GC-MS Profiles

According to the TLC profile, the fraction 2 of Ageratum showed three dark spots at the top site of the TLC plate when it was detected under UV_254_ (see Figure 7a,b). There were two other spots with a green color also detected under UV_254_ and visible light. Under UV_365_, all spots were detected having fluorescence, therefore, it could be assumed that the compounds should have chromophore and a long auxochrome aside with a rigid structure. On the other hand, the fraction 1 of Ixora contains six spots under UV_254_ (see Figure 7c,d). Of these, one was a dark spot and the other five green spots, which could be associated to the chromophore and a long auxochrome. In contrast, the dark spot was undetectable under UV_365_, which could be due to the non-rigid structure. The green spots were detected as a pink fluorescence under UV_365,_ which could be associated with its rigid structure.

The GC chromatogram of Ageratum fraction (fraction 2) and its mass spectra are presented in Figure 8 and Figure 9, respectively. The fraction 2 was separated into four peaks having R_t_ as followed: Rt 8.825 min, 11.255 min, 12.175 min, and 14.460 min. These peaks were detected as compounds with the mass/ion 522, 538, 543, and 539, respectively. On the other hand, the fraction of Ixora (fraction 1) was separated into three peaks, detected at R_t_ 10.709 min, 12.380 min, and 14.153 min for their respective mass/ion 548, 529, and 528 (see Appendix A).

## 3. Discussion

Molecular docking has been applied successfully to discover potent compounds from natural products for various diseases such as neuraminidase inhibitors [32], dengue protease inhibitors [33], and cancers [34]. In the current study, in silico screening using molecular docking has been applied to search for natural product compounds that have the potentials to be developed as a chemotherapeutic agent for breast cancer through PEX9 inhibition.

In the assay system, the extract was added into the sample in order to prevent the binding of the peptide substrate into the enzyme. Peptide substrate has a fluorophore that is connected by an amide link. When the enzyme cleaves it, the fluorophore is released and fluorescence is detected by the UV detector. The presence of the crude extract supposedly reduces the fluorescence reading as the activity of the enzyme is inhibited. The positive control used in this study is NNGH, which is a hydroxamate compound known to bind to MMP at the catalytic site. This catalytic site is composed by zinc ion coordinated by histidine triad (401, 405, and 411) and essential residue of GLU402 [35]. The hydroxamate group will form a chelate with the zinc ion as well as GLU402 to attract water which is closer to scissile amide bond of the inhibitor, thus preventing the proteolysis of the peptide substrate by the enzyme.

The six of eight plants inferred from the in silico screening results showed significant inhibition to MMP9. The fact that the protein model used in in silico screening is the hemopexin domain of MMP9, therefore, the activities of the eight extracts are postulated to inhibit the PEX9 domain. Based on the results of the in vitro MMP inhibition assay, extracts that showed IC_50_ ≤ 100 µg/mL were further selected for the cytotoxicity assay to investigate the inhibition on the 4T1 cells proliferation. According to National Cancer Institute (NCI, Bethesda, MD, USA), the level of crude extract’s inhibition with IC_50_ not greater than 20 µg/mL is categorized as active [36]. As observed, the four extracts showed the IC_50_ at the range being considered as active to moderately active. This corresponds to the results of the in vitro study against MMP9, especially for *Ageratum conyzoides* where the extract, partitions, and fraction demonstrate quite good activities in inhibiting MMP9. Furthermore, the extract also actively inhibited the proliferation of 4T1 metastatic breast cancer cells. All Ageratum samples could be suggested to prevent the peptide substrate binding in the catalytic site of MMP9 by interrupting the homodimerization of PEX9 [17]. However, further investigation is still warranted to confirm the activity of the extract on the cell migration assay. The selectivity of the extracts toward MMP9 and MMP2 should be performed because both have homology in their catalytic domain but non-homology in their PEX domain. The extract that inhibits MMP9 activity better than MMP2 could be associated to having selective inhibition toward MMP9. If the extract activity is about same toward both enzyme, it could be non-selective inhibition that leads to adverse side effect.

GC-analysis and the mass spectroscopic analysis of Ageratum fraction shows a few compounds with *m/z* 522–543 which has not been reported elsewhere [37,38,39,40,41]. The absence of the hit compound of Ageratum namely sesamin with *m/z* 354 in the MS result means that there are some different compounds responsible for the MMP9 inhibition. As such, the MS results of Ixora fraction do not exactly determine the presence of its hit (ixorapeptide I; MR 500.6 g/mol) [42] as predicted by in silico screening. However, the *m/z* of the compounds range from 528–548 assuming the compounds could be a modified ixorapeptide.

According to the cytotoxicity assay, the Ageratum extract tends to be more selective to a triple-negative breast cancer cell than luminal A, because the IC_50_ of the respecting extract is lower when it was treated with 4T1 cell than with T47D cells. This shows a stronger interaction of the extract with the MMP9 protease enzyme than either with the estrogen receptor or progesterone receptor which is commonly present in luminal A cancer type. In contrast, doxorubicin was more potent to T47D (IC_50_ 9 µg/mL) than to 4T1 (IC_50_ 37 µg/mL). This could be the reason why the activity of Ageratum extract is stronger in 4T1 than T47D. Furthermore, the SI of Ageratum extract against 4T1 (14.27) is considerably high and even better than doxorubicin (SI 5.71), therefore the activity of the Ageratum extract is associated to its selectivity rather than cytotoxicity which is good for anticancer drug.

Previous study reported that Ixora flower aqueous extract demonstrated antiproliferative activity with IC_50_ of 31.3 µg/mL toward human breast adenocarcinoma cells [43]. Another study reported that Ixora flower ethanolic extract can reduce the proliferation of human prostate cancer cell line (LNCAP) with IC_50_ 233.9 µg/mL without causing toxicities [44]. Prabhu et al. (2018) supported the previous finding by reporting the antiproliferative activity against Dalton’s lymphoma ascetic (DLA) cells and Erlich Ascites Carcinoma (EAC) cell lines. The flowers were extracted using petroleum ether and methanol extract, and the result showed that the methanol extract performed the best activity with IC_50_ 250 µg/ mL and 300 µg/mL against DLA and EAC, respectively [45]. A recent study of Ixora antiproliferative activity was performed against three different human cancer cell lines, i.e., uterine cervical (HeLa), lung (NCI H-460) and breast (MCF-7) cancer cell lines. The chloroform extract of this flower exhibited IC_50_ as follows: 15 µg/ mL, 3.8 µg/ mL, and 230 µg/ mL toward HeLa, NCI H-460, and MCF-7, respectively [46].

On the other hand, an ethylacetate extract of Ageratum leaves exhibited the cytotoxic activity on adenocarcinomic human alveolar basal epithelial cells (A-549) and leukemia cancer cells (P-388) with IC_50_ values of 0.68 and 0.0003 μg/mL, respectively [47]. To date, there have been not so many cancer cytotoxic study on Ageratum, but more recent research reported that the leaf aqueous extract was reported to inhibit the proliferation activity of leukemic (Jurkat) cells with IC_50_ 408.15 µg/mL and non-toxic activity toward a normal prostate cell lines (PNT2) [48].

Presently, the antiproliferative activity of Ixora and Ageratum contributes significantly to anticancer drugs than the traditional herbal sources as shown in previous studies, especially for the investigation of potential anticancer effect through molecular mechanism of MMP9 in the triple negative cancer type. This sharpens our hypothesis to the conclusion that both Ixora and Ageratum have cancer cell antiproliferative activity, which is due to the MMP9 inhibition, leading to more selective cancer chemotherapy. Attention is paid more to Ageratum than Ixora, as its methanol extract showed IC_50_ 22 µg/mL with SI 14.27 against metastatic cancer line, which is closer to the expected drug-like properties than the previously reported. To the best of our knowledge, the activity of methanol extract, its partition (*n*-hexane, ethylacetate, *n*-butanol) and the *n*-hexane: ethylacetate fractions of both herbs could be potential for novel triple negative anticancer agents from herbal sources, which is not only effective to the cancer cells but also selective toward the normal cells.

## 4. Materials and Methods

### 4.1. Software and Hardware

PEX9, PDBID: 1ITV [49] was downloaded from the Protein Databank (PDB, www.rcsb.org), and the structures of known PEX9 inhibitors (external validation data) that have arylamide structure linked to the planar pyrimidine ring by flexible ethylene chain were taken from published literatures [24,25]. We created an in-house natural compound database containing 200 compounds (ligands). The information on natural compounds was collected from Indonesia’s Herbal Remedies Database (http://herbaldb.farmasi.ui.ac.id), Nature Based Discovery (NADI) System [32] and Natural Product Activity and Species Source Database (http://bidd2.nus.edu.sg) [50]. The 3D structures’ files of the natural compounds were then downloaded from PubChem (https://pubchem.ncbi.nlm.nih.gov/compound). The in silico screening used molecular docking protocol in AutoDock Vina (autodock.scripps.edu) and the output was visualized using Discovery Studio 3.5 (www.accelrys.com). HP laptop with Core i3 processor on Windows 10 operating system with 4 GB RAM and 500 GB Hard Disk was the hardware.

### 4.2. Chemicals

Methanol and other organic solvents, silica for column chromatography, and thin layer chromatography (TLC) F_254_ plates with analytical grade were purchased from Merck, Darmstadt, Germany. The MMP9 enzyme kit was obtained from BioVision comprised of lyophilized MMP9, FRET-based MMP9 substrate (Mca-Pro-Leu-Gly-Leu-Dpa-Ala-Arg), MMP9 assay buffer, and NNGH inhibitor (*N*-isobutyl-*N*-(4-methoxyphenylsulfonyl)-glycyl hydroxamic acid) as its positive control. 4T1 cells (American Type Culture Collection, Manassas, VA, USA), were cultured in RPMI-1640 medium containing 10% (*v/v*) fetal bovine serum (FBS) and 1% penicillin–streptomycin (Life Technologies, Carlsbad, CA, USA) at 37 °C in 5% CO_2_ humidified incubator. T47D cells and RNase were courtesy from Parasitology Laboratory, Medical Faculty, Gadjah Mada University cultured in Dulbecco’s Modified Eagle Media (DMEM). Doxorubicin (DOX) and 3-(4,5-dimethylthiazol-zyl)-2,5-diphenyl tetrazolium bromide (MTT) were obtained from Sigma, St. Louis, MO, USA.

### 4.3. Control Docking and External Validation

The crystal structure of PEX9 with sulfate ion was downloaded from the Protein Data Bank (PDB) with PDBID 1ITV. This ion was used as the native ligand and found located at blade 3 and blade 4 of the pocket site. PEX9 was presented as homo-dimer in the crystal structure and in this study, only one monomer was used in the modeling. The sulfate ion was separated from PEX9 using Discovery Studio 3.5, saved as PDB file and then assigned with Gasteiger Charges using AutoDockTools1.5.6 [51]. PEX9 was prepared using the same program whereby polar hydrogens were retained and the molecule was assigned with Kollman charges. The grid box was automatically defined by PyRx program (exhaustiveness = 32; size 25, 25, 25 and center x = −42.05, y = −30.85, z = −7.26) and the docking was run using AutoDock Vina embedded in PyRx program [52]. The docking parameter was defined as valid, provided that the RMSD values of the complex to be less than 2 Å [27]. Numerous compounds were published as inhibitors for PEX9 [28,29] and they could be used for the external validation before the docking parameters were used for in silico screening.

### 4.4. In Silico Screening

The structures of the natural product compounds (200 ligands) were downloaded from PubChem and then converted to PDB file using Discovery Studio 3.5. The virtual screening was later done using the same protocol as that of the control docking. The output was then collected as csv file and the compounds were tabulated according to the binding affinity. Twenty compounds with the lowest binding affinity were shortlisted as the in silico hits. The in silico hit compounds then were linked to the source plants of which they were reported to have been isolated. Based on the availability, eight of the 20 plants hits were subjected to methanol extraction for in vitro test against MMP9.

### 4.5. Plant Collections and Extractions

Eight selected plants were collected from Specific Region of Yogyakarta, Sukoharjo District, and Semarang City located in the Java Island of Indonesia. Plants’ identification was carried out by referring Flora of Java [53] and further authenticated by Dr. Chusnul Chotimah, Taxonomist of Herbal Materia Medica, Batu, Malang, Indonesia. The voucher specimens were deposited in the Laboratory of Herbal Materia Medica, Batu, Malang, Indonesia with specimen numbers as followed: *Hibiscus rosa-sinensis* (BMM/0013/36-XI), *Ageratum conyzoides* (BMM/0013/37-XI), *Amaranthus spinosus* (BMM/0013/38-XI), *Cordyline fruticosa* (BMM/0013/39-XI), *Ixora coccinea* (BMM/0013/40-XI), *Melaleuca leucadendron* (BMM/0013/41-XI), and *Turnera diffusa* (BMM/0013/42-XI).

### 4.6. Liquid-Liquid Partitions and Fractionation

Total of 40.0 g of extract was put in separating funnel and 800 mL of distilled water was added. Around 800 mL of *n*-hexane was added into the solution and vigorously shaken until dual phase was formed. The *n*-hexane phase was collected and then the water phase was re-shaken with ethyl acetate with the same volume until a clear part appeared in ethyl acetate phase. The obtained *n*-hexane and ethyl acetate phases were then evaporated under reduced pressure to collect ethylacetate partition. The water phase was further partitioned using *n*-butanol while leaving it as the last partition. The selected partition based on in vitro assay was subjected for fractionation using column chromatography. On one hand, the ethylacetate partition of Ixora (5.0 g) was poured into column packed up with silica. The mobile phase containing *n*-hexane: ethyl acetate (3:1) was flowed down through the column while carrying the compound contained in the partition. On the other hand, the *n*-hexane partition of Ageratum (1.0 g) was eluted using *n*-hexane: ethyl acetate (5:3). The fractions were collected and then combined, based on the similar TLC profile. The most collected fraction was then characterized by its chromatography-mass profile using GC-MS (QP2010S Shimadzu).

### 4.7. In Vitro MMP9 Inhibition Assay

The lyophilized enzyme was reconstituted using 110 µL of glycerol 30% in deionized water. Then, the enzyme was diluted into 550 µL of buffer and was ready to be used in the assay. The sample (crude extract/ partitions/ fractions) for the assay was prepared by dissolving it in DMSO to yield the final concentration of 1 mg/mL in a 96-microwell plate. The final concentration of DMSO in the well plate was 0.1%. Briefly, the samples were properly mixed with the buffer before adding the enzyme. The mixture was then incubated at 37 °C for 30 min followed by addition of the substrate (40 µM) and then continued incubating at 37 °C for 60 min. The fluorescence was read using Synergy HTX-3 Multi-mode Reader at 325/393 nm. NNGH inhibitor was prepared using a similar process except for the final concentration set at 40 µM. Four extracts where the % inhibition reached 50%, were then subjected to IC_50_ determination by preparing them in a set of concentrations (0.125 mg/mL; 0.250 mg/mL; 0.500 mg/mL; and 1 mg/mL). Data calculation and the drug-dose dependent curve were prepared by using GraphPad Prism 5 version 5.01 (www.graphpad.com).

### 4.8. In Vitro Cytotoxicity Assay

To determine the cytotoxicity of each extract on 4T1, T47D, and Vero cells, the MTT assay was employed. 4T1 is a triple-negative breast cancer cell line from *Mus musculus* [54] that contains PEX9 showing 61% homology with PDB 1ITV upon Blast Analysis [55]. T47D is a cell model to characterize the progesterone-specific effect of a luminal A subtype of breast cancer [56], whereas Vero cell is a non-tumorigenic cell from the kidney tissue of African green monkey [57]. The assay detects the reduction of MTT and reflects the normal functioning of mitochondria and hence cytotoxicity. Cells (1 × 10^4^/well) were seeded in 96-well flat-bottomed plates and incubated with each extract at various concentrations for 24 h. Doxorubicin was added to the cultures at the following concentrations: 0, 2.5, 5, 10, 20, and 40 µg in a final volume of 100 µL as a positive control using DMSO 0.1%. Extract solutions were prepared in the following concentrations: 10, 20, 40, 80, 160, 320, 640, and 1280 µg/mL. Total of 30 µL of MTT solution (5 mg/mL in PBS) was added to each well and the plate was incubated at 37 °C for another 4 h. Then, the medium was discarded and 150 µL of DMSO was added to dissolve the formazan crystals. The absorbance of each sample was read at 595 nm using a microplate reader. Results were expressed as a percentage of cell viability with respect to untreated control cells (as 100%) [58]. The data calculation and the drug-dose dependent curve were prepared by using GraphPad Prism Version 5.01. Morphological studies were carried out by visual observation through microscopy (Center Valley, PA, USA). 4T1 Cell were stained sequentially with 3 mΜ CellEvent™, Mitotracker RedCMXRos (Invitrogen, Carlsbad, CA, USA), and Hoechst 33342 (Invitrogen) at a cell concentration of 105 cells/mL in the dark room for 30 min at 37 °C. Cell morphology was observed using the Olympus FluoView FV1000 confocal laser scanning microscope (Center Valley, PA, USA) with a 60X objective and further analyzed using the FV1000 software. Viable cells appeared as leaf-shape and appear white nodes, while dead cells as round and appear black node.

### 4.9. Gas Chromatography-Mass Spectroscopy

Total of ± 5 mg of individual fraction 1 (Ixora) and fraction 2 (Ageratum) were dissolved in 1 mL of chloroform in separated vials. From each vial, 0.5 μL of solution was injected into GC-MS machine with Rtx 5 MS column (diphenyldiethylpolysiloxane) as the column. The GC is coupled to a mass spectrometer (Shimadzu QP2010SE, Kyoto, Japan), which has the onus function of recording the mass spectrum of the chemical compounds as they come out of the GC and after fragmentation processes by a stream of electrons in the mass spec. Helium gas was used a carrier gas, and the GC oven was initially held at 100 °C for 5 min, and then elevated 5 °C per min until reaching 300 °C. Peaks in the chromatograms produced by these analyses were identified by a combination of references to their mass spectra and the NIST08 mass spectral database.

## 5. Conclusions

This study provides evidence that in silico study can aid and accelerate the drug discovery process from natural product. The results also suggest that the aerial part of *Ageratum conyzoides* has the potential to be developed as a herbal remedy for breast cancer. However, further investigation is still needed to confirm its cytotoxicity in a normal human cell and to isolate and elucidate the compounds that are responsible for the activity at the PEX9 domain of MMP9.

## Figures and Tables

**Figure 1 molecules-25-04691-f001:**
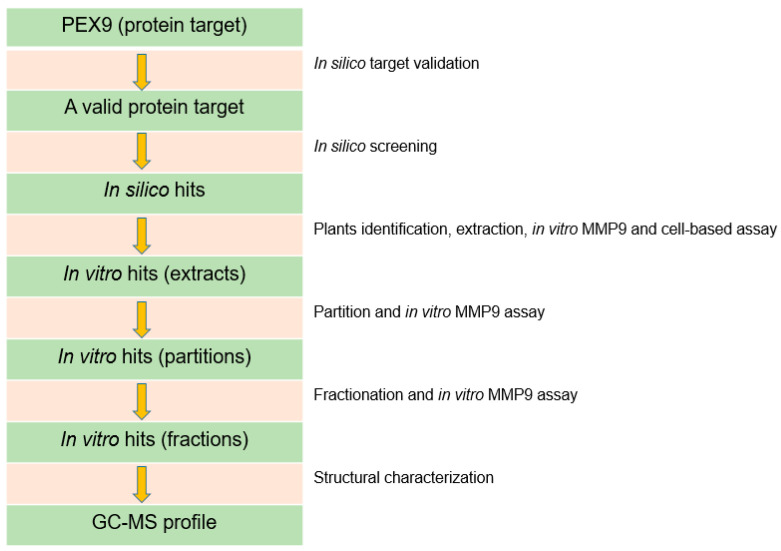
The flow chart of the studies in bioguided fractionation of local plants to identify MMP9 inhibitors and breast cancer cytotoxic agents.

**Figure 2 molecules-25-04691-f002:**
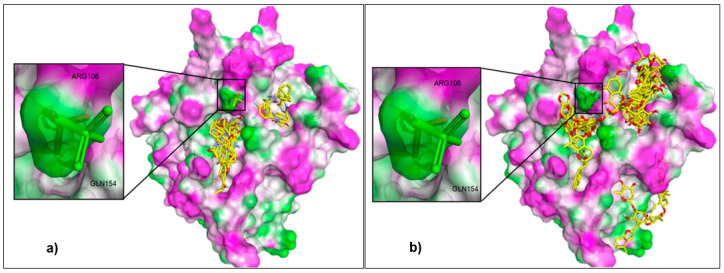
Superimposition of (**a**) 17 published PEX9 inhibitor and (**b**) top twenty ligand scores docked into PEX9 with deep pocket in the left side and shallow pocket in the right side. Inset is the control docking of sulfate ion with green stick is initiate pose and red stick is re-docked pose. The external ligands are colored in stick model with C = yellow; H = white; O = red; and N = blue. The protein is presented in a surfaced form with magenta area = hydrogen bond donor; white = neutral and green = hydrogen bond acceptor.

**Figure 3 molecules-25-04691-f003:**
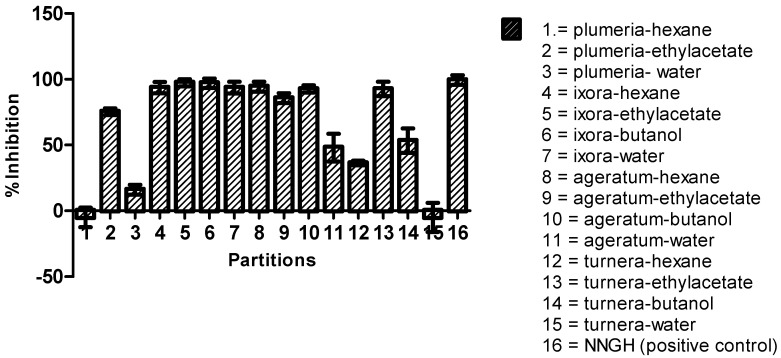
The histogram of inhibition percentage from 15 partitions from methanol extract of each plant measured by FRET-based assay against MMP9 activity in vitro. *N*-isobutyl-*N*-(4-methoxyphenylsulfonyl)-glycyl hydroxamic acid (NNGH) was used as the positive control by showing 100% inhibition at 1 µM. The bars described the standard error of triplicate assay.

**Figure 4 molecules-25-04691-f004:**
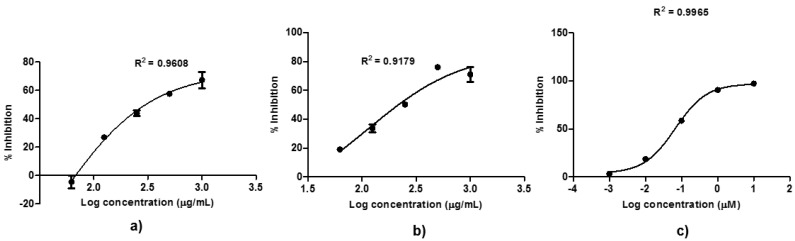
The drug dose-dependent curves of two active fractions, i.e., (**a**) Ageratum aerial parts, and (**b**) Ixora leaves and (**c**) NNGH as the results of FRET-based assay toward MMP9. NNGH was used as the positive control by showing IC_50_ 47.8 nM. The bars showed the standard error of triplicate assay as previously described.

**Figure 5 molecules-25-04691-f005:**
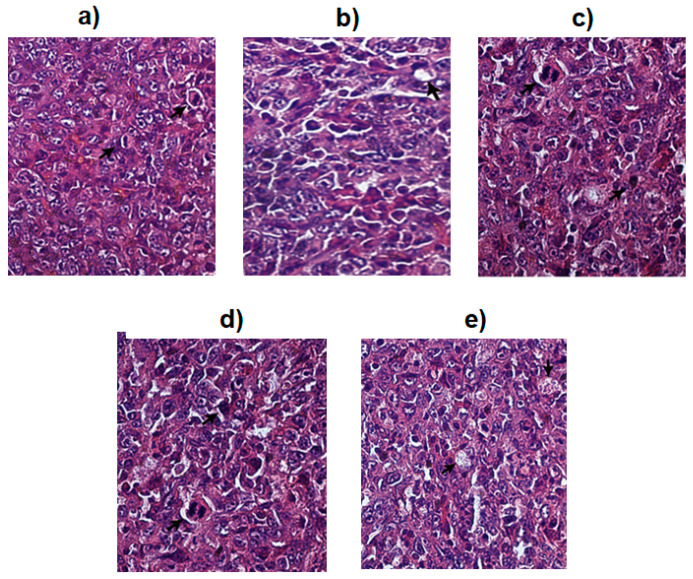
The 4T1 cell morphologies after and before treating with four top active extracts as followed; (**a**) untreated cell, (**b**) Ixora leaves, (**c**) Turnera leaves, (**d**) Ageratum aerial part, and (**e**) Plumeria leaves. The arrow indicates the living cell for the white nodes whereas the death cell is black node surrounded by white line.

**Figure 6 molecules-25-04691-f006:**
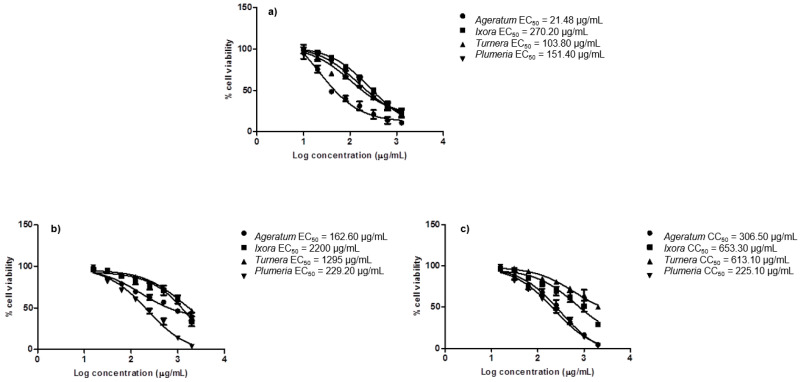
The drug dose-dependent curves of four top active extracts against (**a**) 4T1, (**b**) T47D, and (**c**) Vero cells.

**Figure 7 molecules-25-04691-f007:**
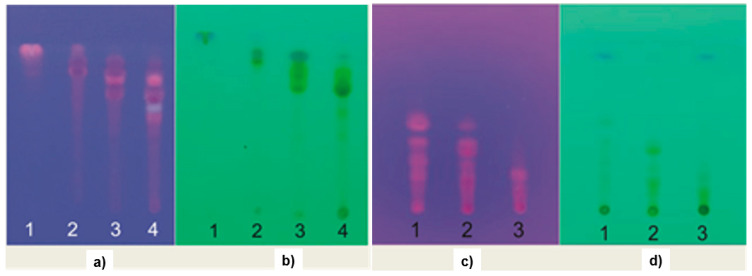
The TLC chromatogram of the fraction 1–4 of Ageratum which are detected under (**a**) UV_365_ and (**b**) UV_254_. The fraction 1–3 of Ixora are indicated under (**c**) UV_365_ and (**d**) UV_254_.

**Figure 8 molecules-25-04691-f008:**
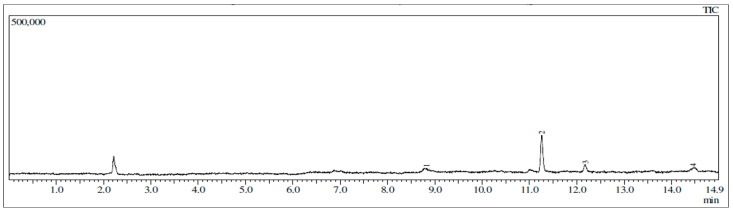
The GC chromatogram of Ageratum active fraction showing four peaks in a different retention time. The peak at 2.250 min is of chloroform which is used as solvent.

**Figure 9 molecules-25-04691-f009:**
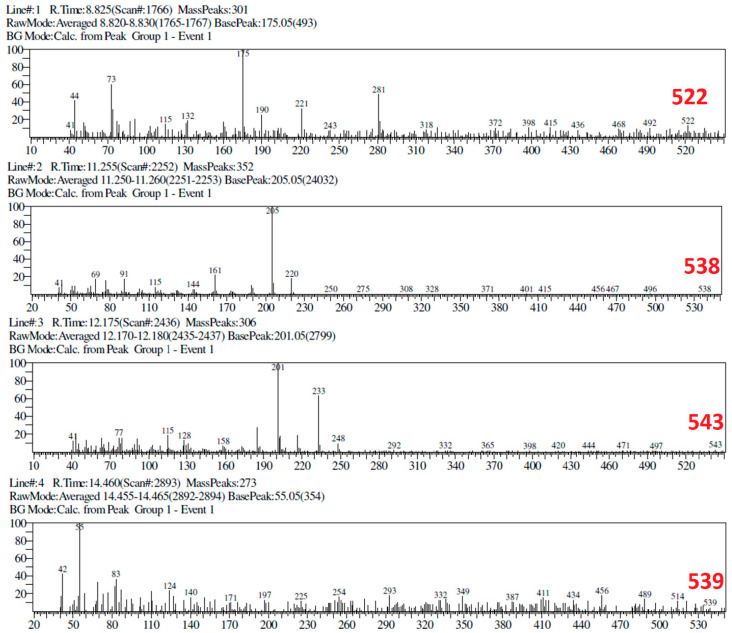
The mass spectra of four peaks identified from GC chromatogram of Ageratum fraction 2. According to Figure 8, four peaks with R_t_ 8.825 min, 11.255 min, 12.175 min, and 14.460 min are detected as compounds having mass/ ion: 522 (base peak 175), 538 (base peak 205), 543 (base peak 201), and 539 (base peak 55), respectively. The base peak informs the most stable fragment during electron impact in MS characterization.

**Table 1 molecules-25-04691-t001:** The binding affinity of the top 20 ligands from the herbal database with their corresponding source plant.

PubChem ID	Ligand	Binding Affinity (kcal/mol)	Plant
441207	digitoxin	−11.2	*Cordyline fruticosa*
2724385	digoxin	−10.4	*Cordyline fruticosa*
91540	gitoxin	−10.1	*Cordyline fruticosa*
10328286	thonningianin A	−9.1	*Piper betle*
6325284	amaranthin	−8.7	*Amaranthus spinosus*
4483248	ophiopogonin C	−8.6	*Parkia javanica*
92825	gypsogenin	−8.6	*Turnera diffusa*
101277	pachyrrizin	−8.5	*Pachyrhizus erosus*
5484010	sequoiaflavone	−8.4	*Elateriospermum tapos*
11467	γ-terpineol	−8.4	*Melaleuca leucadendron*
15411208	2,3-dihydrowithaferin A	−8.4	*Withania somnifera*
72307	sesamin	−8.3	*Ageratum conyzoides*
100257	Thalrugosin	−8.3	*Cyclea barbata*
15484640	ixorapeptide I	−8.3	*Ixora coccinea*
23265223	quercetin−3-*O*-(3′6″-*O*-di-*p*-coumaroyl)-glucoside	−8.2	*Hibiscus rosa-sinensis*
131750919	Cryptochrome	−8.2	*Averrhoa carambola*
92097	Taraxerol	−8.2	*Plumeria alba*
99620	Homoaromoline	−8.2	*Arcangelisia flava*
162807	(−)-glyceollin I	−8.1	*Glycine soja*
490367	19-α-hydroxyasiatic acid	−8.1	*Cordyline fruticosa*

**Table 2 molecules-25-04691-t002:** The extract yields from eight selected plants and their % inhibition against MMP9 in vitro at 1 mg/mL alongside with their IC_50_.

Crude Extract	Yields (%)	% Inhibition ± SEM	IC_50_ ± SEM (µg/mL)	R^2^
*Cordyline fruticosa* leaves	43	−6 ± 12	ND	ND
*Amaranthus spinosus* aerial part	62	81 ± 4	783 ± 40	0.7501
*Turnera diffusa* leaves	65	55 ± 4	495 ± 20	0.9898
*Ageratum conyzoides* aerial part	65	75 ± 3	64 ± 14	0.8071
*Ixora coccinea* leaves	81	86 ± 1	82 ± 3	0.9879
*Hibiscus rosa-sinensis* leaves	72	55 ± 2	822 ± 20	0.9723
*Plumeria alba* leaves	77	85 ± 9	24 ± 8	0.8268
*Melaleuca leucadendron* leaves	41	−15 ± 14	ND	ND

ND = not determined.

**Table 3 molecules-25-04691-t003:** The results of in vitro cytotoxicity assay of four extracts against 4T1 and T47D cell growth. IC_50_ here describes the concentration of the extract that inhibits at least 50% cell proliferation, whereas the safety index (SI) is the ratio between IC_50_ of cancer cells (4T1 and T47D) with the IC_50_ of Vero cells.

No	Sample	IC_50_ ± SEM (µg/mL)			SI	
		4T1 (R^2^)	T47D (R^2^)	Vero (R^2^)	4T1	T47D
1	*Turnera diffusa* leaves	104 ± 5 (0.9192)	1295 ± 9 (0.9680)	613 ± 6 (0.9298)	5.90	0.47
2	*Ageratum conyzoides* aerial part	22 ± 14 (0.9388)	163 ± 3 (0.9387)	307 ± 5 (0.9708)	14.27	1.88
3	*Ixora coccinea* leaves	270 ± 3 (0.9887)	2200 ± 42 (0.8831)	653 ± 10 (0.9058)	2.42	0.29
4	*Plumeria alba* leaves	151 ± 4 (0.9566)	229 ± 4 (0.9741)	225 ± 4 (0.9741)	1.49	0.98
5	Doxorubicin	37 ± 6 (0.9483)	9 ± 4 (0.8740)	211 ± 7 (0.9549)	5.71	24.59

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
