# Peer review of "Bioguided Fractionation of Local Plants against Matrix Metalloproteinase9 and Its Cytotoxicity against Breast Cancer Cell Models: In Silico and In Vitro Study"

_molecules, 2020, doi:10.3390/molecules25204691_

Round 1
Reviewer 1 Report
Comments:
Line 29: Could you, please, specify the general terms "partitions" and "fractions" in regard to the experimental procedures followed in the present study?
Line 46: better to omit the phrase "also called as Claudin-Low subtype"(see, e.g., Fougner et al., Re-definition of claudin-low as a breast cancer phenotype, Nature Communications, 2020)
Lines 108-116: I probably miss something here: If all 17 compounds have been predicted to be active in terms of their calculated binding affinity (in all cases: <-3.5 kcal/mol), then comparison of calculated binding affinity with the corresponding Kd value ("true" activity value) may result in just two groups: either true positive (TP) or false positive (FP). Could, you, please, explain further classification in true negative (TN) and false negative (FN) groups?
Lines 241-247 (3.5. GC-MS profiles): The paragraph is difficult to follow and has to be re-written in a clearer way (e.g. lines 253-254: "Furthermore, under UV365, the spot having dark color was undetectable associating with no rigid structure to perform that such fluorescence"). For clarity reasons, Figure S1 and Figure S2 have to be mentioned in this paragraph, too. Moreover, it might be better if the above Figures were incorporated in the main manuscript.
Discussion: The text devoted to biological hypotheses (e.g. lines 292-301) may be shortened. On the other hand, Discussion should be enriched focusing on the novelty of the findings and further supporing the Conclusion "this study provides evidence that in silico study can aid and accelerate the drug dicovery process from natural product".
Reviewer 2 Report
- Reference 1 is too old. Please update the first sentence and the reference.
- Describe RMSD when first mentioned (line 99). All abbreviated words need to be spelled out.
- Describe Kd when first mentioned and note how the values were calculated.
- Figure 3, its caption and the related text (lines 176-180): please describe what assay has been performed to test the inhibitory action. Complete the caption indicating the number of replicates, what are the error bars showing, was there any positive control for the inhibitory action, were there any significant differences there?
- Figure 4: Complete the caption as requested above.
- Line 426: at what wavelength?
- GraphPad Prism 5, which version?
- In the methods, there is no section describing the morphology studies? What staining was used and how to confirm which color indicates live cells or dead cells?
- Describe IC50 and EC50 when first mentioned in the text. Include R squared for the calculated IC50 values.
- It seems to me that you are talking about inhibitory concentrations so IC50 makes more sense than EC50. Be consistent.
- Table 3: the captino is so incomplete. Define EC50, CC50 and SI.
- Table 4: what are the results on the normal cell line?
- The discussion needs to be more enriched than simply summarizing the results. Similar studies which have investigated these plants are recommended to be included in the discussion. For example the following papers (but not limited to these papers):
In vitro Anti-Cancer Study of Vitis Viniferae, Ixora Coccinea and Piper Longum Ethanolic extracts on Human Breast Carcinoma Cells
Anticancer activity of Ixora coccinea linn flower extracts against Human Breast Adenocarcinoma cells
Cytotoxic activity of extracts of Ixora species and their GC-MS analysis
- Also, for interested readers, the introduction could be improved talking about importance of natural medicine as a source of treatment for cancer and breast cancer, as so many studies are on going. Here are some examples:
Bacopasides I and II Act in Synergy to Inhibit the Growth, Migration and Invasion of Breast Cancer Cell Lines
Elaeagnus angustifolia Plant Extract Inhibits Epithelial-Mesenchymal Transition and Induces Apoptosis via HER2 Inactivation and JNK Pathway in HER2-Positive Breast Cancer Cells
Anthocyanins Isolated from Vitis coignetiae Pulliat Enhances Cisplatin Sensitivity in MCF-7 Human Breast Cancer Cells through Inhibition of Akt and NF-κB Activation
- What are the limitations of this study?
- The text needs to be polished for minor English errors.
Round 2
Reviewer 1 Report
General comments on Results and Discussion
The main finding of this in silico and in vitro work, i.e. high anti-cancer activity along with a high safety index of a specific fraction (fraction 2) obtained from Ageratum conyzoides, is more clearly presented in the revised manuscript. On the other hand, much more work is necessary before fully identifying the specific ingredient(s) of the aforementioned active fraction with anti-cancer activity. Relevant results shown in the present study, especially in Figure S1 and Figure 9, should be further explained and commented on by the authors (or, omitted).
Specific comments
Line 128: please, add the following clarification: "...three may be considered as "marginally active" (1.00 μM<Kd<1.50 μM) and another three are inactive". Otherwise, the argument "...the TPR was equal to 82%..." (line 139) is not valid (11:17 x 100 = 65%)
Line 211: Table S2 (instead of Table S3)
Line 235 (Table 3): The words "Line 223-225. Table 3" should be omitted.
Line 270: TLC and GC-MS profiles (instead of GC-MS and TLC profiles)
Lines 282-282 (Figure 7): c and d are not shown on the Figure.
Lines 286-291: is there any possible relationship between the TLC spots of fraction 2 (Ageratum) and fraction 1 (Ixora) and the corresponding GC peaks? Please, add any comments.
Line 294: Figure 8 (instead of Figure 7)
Line 298: Figure 9 (instead of Figure 8)
Lines 298-300: The legend of the Figure has to change and be more explanatory.
Lines 499-503: Please, provide more technical details.
Line 500: "±5 mg of fractions..." is something missing?
Author Response
Dear Reviewers,
Thank you very much for thoroughly checking and criticizing our manuscript that would be a great value for us to increase our quality in writing as well as to guide our future research.
Herewith we submit the minor revision version of our manuscript entitled " Bioguided Fractionation of Local Plants against Matrix Metalloproteinase9 and Its Cytotoxicity against Breast Cancer Cell Model: In Silico and In Vitro Study" Molecule-953688 to be considered for publication in Molecule.
We attach a point-by-point response based on the reviewers' comments.
We hope very much that our manuscript meets your approvals.
Kind regards
Maywan Hariono, Ph.D.
Faculty of Pharmacy, Sanata Dharma University
Indonesia

This manuscript is a resubmission of an earlier submission. The following is a list of the peer review reports and author responses from that submission.